# Expanding the Hepatitis E Virus Toolbox: Selectable Replicons and Recombinant Reporter Genomes

**DOI:** 10.3390/v15040869

**Published:** 2023-03-28

**Authors:** Noémie Oechslin, Maliki Ankavay, Darius Moradpour, Jérôme Gouttenoire

**Affiliations:** Division of Gastroenterology and Hepatology, Lausanne University Hospital and University of Lausanne, 1011 Lausanne, Switzerland

**Keywords:** epitope tag, infectious clone, insertion, reporter, selection gene, subgenomic replicon

## Abstract

Hepatitis E virus (HEV) has received relatively little attention for decades although it is now considered as one of the most frequent causes of acute hepatitis worldwide. Our knowledge of this enterically-transmitted, positive-strand RNA virus and its life cycle remains scarce but research on HEV has gained momentum more recently. Indeed, advances in the molecular virology of hepatitis E, including the establishment of subgenomic replicons and infectious molecular clones, now allow study of the entire viral life cycle and to explore host factors required for productive infection. Here, we provide an overview on currently available systems, with an emphasis on selectable replicons and recombinant reporter genomes. Furthermore, we discuss the challenges in developing new systems which should enable to further investigate this widely distributed and important pathogen.

## 1. Introduction

The hepatitis E virus (HEV) was discovered in 1983 as a causative agent of acute hepatitis among Soviet soldiers in Afghanistan [1] and has been molecularly cloned in 1990 [2]. It has been classified as the sole member of the *Hepeviridae* family. With advanced sequencing technologies and exploration of samples from diverse origins, this family expanded rapidly and is now divided into two subfamilies, the *Orthohepevirinae* and the *Parahepevirinae*, of which the latter comprises only the *Piscihepevirus* genus, corresponding to viruses infecting fish [3] (taxonomy available at https://ictv.global/report/chapter/hepeviridae/hepeviridae (accessed on 1 March 2023)). The subfamily *Orthohepevirinae* comprises the genera *Avihepevirus* (bird viruses), *Chirohepevirus* (bat viruses), *Rocahepevirus* (rodent viruses), and *Paslahepevirus* (mammalian viruses). The species *Paslahepevirus balayani* is composed of eight HEV genotypes of which HEV-1 to HEV-4 represent the most important human pathogens (Figure 1A). However, recent observations indicate that infection of humans can also occur by other viruses including camel HEV-7 [4] or the more distantly related rat HEV ([5,6] and refs. therein).

HEV infection of humans causes mostly self-limiting acute hepatitis which can be accompanied by jaundice. HEV-1 and -2, which are transmitted from humans to humans via the fecal–oral route in resource-limited settings with poor sanitation, are highly pathogenic and may lead to severe hepatitis with up to 25% fatality rate in pregnant women [7,8]. Infections with HEV-3 and -4, which represent a zoonosis transmitted primarily through the consumption of undercooked pork or game meat, are often asymptomatic but can cause symptomatic acute hepatitis, especially in middle-aged and elderly men, trigger acute-on-chronic liver failure or be associated with neurologic, renal or other extrahepatic manifestations [9,10]. HEV-3, as well as HEV-3ra, HEV-4, HEV-7, and rat HEV, may persist in immunocompromised patients and cause chronic hepatitis leading to cirrhosis and liver failure [11].

HEV harbors a 7.2-kb positive-strand RNA genome resembling a cellular mRNA with a m7G cap and a poly-A tail (Figure 1B). The genome encodes three main open reading frames (ORF) and, likely, a fourth, in the case of HEV-1 [12]. ORF1, which is first translated upon infection, encodes the so-called replicase, a large, multifunctional protein harboring methyltransferase (MeT), macro, helicase (Hel), and RNA-dependent RNA polymerase (RdRp) domains (Figure 1B). While MeT, Hel, and RdRp have intrinsic roles in viral RNA synthesis, including capping, macro acts as an enzyme known to remove ADP-ribosylation from the posttranslationally modified proteins [13]. A papain-like cysteine protease (PCP) domain has been identified by sequence homology but has not been functionally confirmed yet [14]. A domain with a high-sequence variability known as the hypervariable region (HVR) is located in the middle of ORF1 but has no recognized function. The partially overlapping ORF2 and ORF3, encoded by the last third of the genome, are translated from a subgenomic RNA produced by the viral RdRp (Figure 1B). ORF2 encodes the viral capsid protein which can be produced as infectious, glycosylated, and cleaved forms, the latter supposedly serving as immunologic decoys [15,16]. ORF3 encodes a small, palmitoylated protein which is essential for the secretion of infectious virions as quasi-enveloped particles wrapped in exosomal membranes [17,18].

Not the least due to the limited availability of model systems, knowledge of the HEV life cycle remains relatively scarce. The first functional molecular clones of HEV-1, i.e., the pSGI-HEV(I) [19] and Sar55 clones [20], were described in the early 2000s. Molecular clones of HEV-3 have been described more recently and found to replicate more efficiently in cell culture. They include the p6 clone (Kernow-C1 strain, HEV-3a) [21], the 83-2 clone (HEV83-2-27 strain, HEV-3k) [22], and the pJE03-1760F clone (HEV-3b) [23] (reviewed in [24]) (Figure 1A). An increasing number of molecular clones, including for other genotypes, such as HEV-4 [25], HEV-5 [26], HEV-7 [27], and HEV-8 [28], have been described recently but have been less studied yet.

Studies of viral life cycles have been facilitated by the use of recombinant constructs allowing the expression of a reporter gene. These tools have been key in a number of discoveries, including the receptor used for virus entry or the identification of potent antivirals [29,30]. This strategy has been successfully applied to positive-strand RNA viruses from different families, including, among others, corona- [31,32], alpha- (e.g., Venezuelian equine encephalitis virus [29]), flavi- (reviewed in [33]) and hepaciviruses (hepatitis C virus [34], reviewed in [35]).

Here, we provide an overview on currently available HEV clones, including subgenomic replicons and full-length genomes, which have been engineered to allow selection or reporter expression to study the viral life cycle. Applications and expected values of their use shall be discussed together with their limitations. Furthermore, perspectives and challenges for the development of new HEV constructs shall be addressed.

## 2. Subgenomic Replicons as Tools to Study HEV RNA Replication

Subgenomic replicons, i.e., constructs comprising the genome elements and allowing expression of the viral machinery required for RNA replication without infectious particle production, have been established for numerous positive-, but also negative-strand, RNA viruses (reviewed in [36]). They allow the study of genuine viral RNA replication and enable manipulation under biosafety level 2 conditions. Thus, replicons have been widely used to study pathogenic RNA viruses and were even instrumental for the development of antiviral therapy against HCV [37].

The first subgenomic HEV replicon was described in 2004 by Emerson et al. [38]. A green fluorescent protein (GFP) sequence with two stop codons was inserted after the methionine start codon of ORF2 and replaced part of the ORF2 and most of the ORF3 sequences, while ORF1 was left unmodified. In this setting, the replicon allows for expression of the entire replicase, the first 14 amino acids (aa) of ORF3 and the GFP reporter but no ORF2 protein is produced. While ORF2 expression is not necessary for viral RNA replication, structural RNA elements, namely, internal stem loops 1 and 2 in the middle of its coding region, were shown later to be essential for efficient HEV RNA replication [39]. Therefore, HEV replicons should preserve these structures as well as some others described at the 3′-end of ORF2 [40]. Thanks to this first HEV replicon, it was possible to monitor HEV replication in cell culture by microscopy or fluorescence-activated cell sorting (FACS). The replicon was further improved with the more stable enhanced GFP [41] but, although it represents a powerful tool to identify HEV-replicating cells after replicon transfection, it does not allow for quantitative assessment of viral replication kinetics.

Quantitative evaluation of viral RNA replication can be achieved by the use of a luciferase reporter (Table 1). Replicons harboring the non-secreted Renilla [42,43] or firefly luciferase [44] have been described. However, the most widely used replicon harbors a Gaussia luciferase (Gluc) sequence inserted downstream of the ORF2 start codon, replacing part of its sequence [21]. Gluc possesses a signal sequence resulting in its secretion and accumulation in the cell culture supernatant. It allows the study of sequential time points in the same transfected cell population and, therefore, to follow replication kinetics over several days. The Gluc replicon has served many purposes, including drug screening [45], mutational [46], and domain swapping analyses [47]. Although very convenient, this reporter is limited by the fact that any defect in the secretory pathway may falsely point toward an impaired replication capacity.

Selection of a cell population which autonomously replicates subgenomic HEV RNA is another interest achievable with replicons. Similar to replicons harboring GFP or luciferase reporters, an antibiotic selection cassette, e.g., the neomycin phosphotransferase II gene (Neo), can be inserted at the start codon of ORF2 (Table 1). The selection of cells replicating an HEV-Neo replicon may serve the study of the mechanisms of RNA production [48] or to evaluate antiviral agents [49]. Beyond these applications, selectable replicons allow the performance of reverse genetics studies taking advantage of the intrinsic rapid evolution of RNA viruses and the emergence of adaptive second-site changes. Selectable subgenomic HEV replicons allowed our laboratory to conduct a transposon-mediated random insertion screen to identify viable insertion sites within the ORF1 region [50] and to select for adaptive changes after site-directed mutagenesis of a conserved α-helix in the RdRp, revealing a functional interaction between the thumb and palm subdomains [51]. While selectable replicons can be a useful tool to identify mutations conferring antiviral resistance [52], they have not been used successfully for this purpose in HEV thus far.

**Table 1 viruses-15-00869-t001:** Reporter and selection genes used in HEV replicon constructs. The reference corresponding to the first description of the replicon construct is indicated. BSR, Blasticidin-S resistance; Fluc, Firefly luciferase; GFP, Green fluorescent protein; Gluc, Gaussia luciferase; Neo, Neomycin phosphotransferase II; Rluc, Renilla luciferase; Zeo, Zeocin resistance cassette; and ZsGreen, Zoanthus green fluorescent protein.

Reporter/ Selection Gene	Size (bp)	Application	Ref.
Gluc	555	Replication kinetics	[21]
GFP	720	Evaluation replication level/cell sorting	[38]
Neo	795	Selection of a cell population	[48]
Rluc	932	Replication kinetics	[42]
GFP-Zeo	1077	Selection and eval. replication level	[53]
BSR-2A-ZsGreen	1153	Selection and eval. replication level	[54]
Gluc-2A-Neo	1408	Selection and replication kinetics	[55]
Fluc	1659	Replication kinetics	[44]

Selection markers have also been used in combination with fluorescent proteins, either by direct fusion of the reporters, i.e., GFP-Zeo [53], or by the insertion of the self-cleaving 2A peptide, i.e., BSR-2A-ZsGreen [54] (Table 1). These constructs allow, first, the selection of a population of cells stably replicating HEV and, then, the analysis of the effect of various drugs by evaluating the fluorescence intensity at the single-cell level [53,54]. Similarly, a combination of Gluc with a selection marker has recently been described, i.e., Gluc-2A-Neo, enabling monitoring of viral replication by a simple luciferase activity measurement in the supernatant of a homogenous cell population harboring replicating HEV [55].Interestingly, the strategy consisting in expressing a reporter in place of ORF2 and ORF3 can be applied to different HEV genotypes. Therefore, Gluc or Neo replicons have been prepared, in addition to HEV-1 and -3, for HEV-4 [25], HEV-5 [26], HEV-7 [27], and HEV-8 [28], but also for the more distant rat HEV [56].

In conclusion, the replicon constructs allow for working with a non-infectious system and offer a versatile platform for the incorporation of a wide range of reporters. While this system has contributed to an improved understanding of HEV RNA replication from the virus and the host cell perspective, one should keep in mind that expression of a reporter gene inserted in place of the ORF2 sequence is linked to the synthesis of the subgenomic HEV RNA. Therefore, any mutation which would affect the generation of this RNA species but not of the full-length RNA may result in misinterpretation as a replication defect. In addition, participation of the ORF2 and ORF3 proteins in RNA replication cannot be evaluated. The recent description of an ORF2 single nucleotide variant which may affect viral RNA replication [57] is supporting this concern. Hence, results obtained in a subgenomic replicon system should be validated in a full-length HEV RNA replication and/or infection system.

Trans-complementation systems have also been developed for HEV. In such systems, subgenomic replicons are transfected into cells expressing in trans the ORF2 and/or ORF3 proteins or co-transfected with full-length viral RNA into naïve cells, in order to produce virus-like particles containing the reporter genome or to restore a defective replicase. These systems allow for uncoupling the RNA replication from assembly and genome packaging, which are both still poorly characterized. The first trans-complementation system was developed by transfection of a Gluc replicon into cells stably expressing ORF2 and/or ORF3 [58]. Viral-like particles were successfully produced and secreted into the cell culture supernatant, allowing study of the role of ORF3 protein in virus secretion [58].

A similar system has been established by the same group to study the mechanisms leading to subgenomic RNA synthesis [59]. Here, cells overexpressing wild-type or polymerase-deficient ORF1 protein were transfected with a Gluc construct harboring a deletion in the ORF1 sequence, allowing the mapping of the intragenomic subgenomic promoter [59]. Moreover, a trans-complementation system was also used to package a GFP replicon into viral particles by co-transfection with a full-length RNA genome. In this setting viral particles were infectious, but not all of them delivered a GFP replicon after infection [60]. While trans-complementation assays leading to encapsidation of a replicon construct offer interesting perspectives, such as lowering the biosafety level required for in vitro infection compared to the use of authentic infectious clones, they are strongly hampered by their relative inefficiency in producing virus-like particles. This illustrates the need to improve trans-complementation assays but also to develop full-length HEV genomes harboring reporters.

## 3. Infectious HEV Clones Harboring Reporters

Viruses are under constant selection pressure, forming the basis for their evolution and optimization of genome size. Therefore, viral genomes offer little room to accommodate reporter sequences or entire genes. Moreover, in the case of HEV, available model systems produce relatively low infectious titers as compared to some other viruses. Therefore, identifying viable insertion sites to introduce a reporter gene with minimal impact on replication is challenging. Two approaches can be taken, either by unbiased random insertion and selection for viable insertion sites or by targeting regions of the genome that are less well conserved and, potentially, more tolerant to foreign sequence insertion.

Based on previous work on the hepatitis C virus [61], an unbiased approach to identify viable insertion sites in HEV ORF1 was recently taken by our laboratory. Transposon-mediated random insertion of a 15-nt (5-aa) sequence in a selectable subgenomic replicon (83-2 clone) allowed the identification of viable insertion sites in the MeT, the HVR, and between the Hel and RdRp domains [50]. When the 5-aa transposon sequence was replaced by the larger HA epitope tag, only insertions in the HVR retained full capacity to replicate and to produce infectious virus (Figure 2). These HA-tagged genomes allowed for the convenient and highly reproducible detection of the HEV replicase by immunoblot as well as immunofluorescence, demonstrating that the largely predominant form of ORF1 protein corresponds to the unprocessed polyprotein of ~200 kDa and yielding first insights into putative viral replication sites. Of note, genomes harboring a GFP sequence insertion were found to be replication competent but did not yield any visible fluorescent signal, indicating that reporters of up to 25 kDa may be tolerated in the HVR (unpublished data). Furthermore, we have shown that insertion of NanoLuc flanked by minimal linker sequences is viable (Figure 2). While RNA replication capacity appears reduced as compared to the parental HEV 83-2 clone, the inserted enzyme remained fully functional and allowed for the quantitative assessment of viral replication kinetics. Moreover, a full-length genome harboring NanoLuc in the identified site produced infectious virus, however, at a lower titer than the parental clone [50]. Hence, luciferase activity can serve to quantify viral replication after genuine HEV infection, facilitating future studies on viral entry or antiviral drug discovery efforts.

HEV presents relatively high genetic variability, with ≥25% nucleotide sequence diversity between major human-pathogenic genotypes 1–4. Important intra-genotype variability is also observed, especially in HEV-3, with particularly high sequence diversity in the so-called hypervariable region (HVR) of ORF1 (Figure 2). Of note, HEV-3ra, known to infect rabbits as natural host, usually harbors a highly variable 31-aa (93-nucleotide (nt)) insertion between the macro and helicase domains [62,63,64].

Interestingly, HEV-3 genomes isolated from patients with chronic hepatitis E sometimes harbor insertions of different lengths in the HVR, ranging from 60 to 333 nt ([65,66] and reviewed in [67]). The nature of the inserted sequence is also highly variable, including duplications of HEV sequences from the HVR or RdRp regions or fragments of human mRNA [65,66,68,69,70,71]. Inserted sequences are often rich in prolines as well as polar residues, which is close to the aa composition found in the targeted HVR sequence (Figure 2). Moreover, the well characterized insertion found in the Kernow-C1 p6 infectious clone has been shown to provide a replication advantage in cell culture [21]. This 174-nt (58-aa) insertion derived from ribosomal protein S17 (RPS17) RNA was selected in cell culture after infection with a stool sample from a patient with chronic hepatitis E [69,72]. Hence, the identification of sequence insertions in the HVR of natural HEV isolates suggests that epitope tags or reporter sequences inserted in this region may be viable.

Taking advantage of HEV-3 sequence diversity, Metzger et al. employed a targeted approach to establish full-length Kernow-C1 p6 genomes harboring hemagglutinin (HA, 9 aa) or V5 (14 aa) epitope tags in the HVR [73], at a site where a naturally occurring insertion had been described [74] (Figure 2). Of note, as the aa sequence of the V5 tag comprises 3 proline residues and partly resembles the targeted HVR sequence, the authors adopted a mixed strategy of aa substitution and insertion to prepare this construct. The shorter HA tag was less tolerated at the same position, highlighting the importance of the nature of the inserted aa sequence. In addition to preserved RNA replication, full-length genomes harboring a V5 tag in the HVR retained the ability to produce infectious virus. Metzger et al. exploited functionality of the tagged genomes investigate potential processing and the subcellular localization of the ORF1 protein by immunoprecipitation, immunoblotting and immunofluorescence analyses using epitope-specific antibodies [73].

Similarly, Primadharsini et al. have inserted NanoKAZ luciferase, also known as NanoLuc (171 aa, 19 kDa), in the HVR region of the pJE03-1760F clone [75]. Three of five constructs prepared yielded measurable luciferase activity, and only one was found to be genetically stable upon passaging (Figure 2). The selected HEV genome harboring NanoLuc was able to produce infectious virus, including quasi-enveloped particles, and enabled a drug screen yielding four compounds with potential antiviral activity against HEV [75].

**Figure 2 viruses-15-00869-f002:**
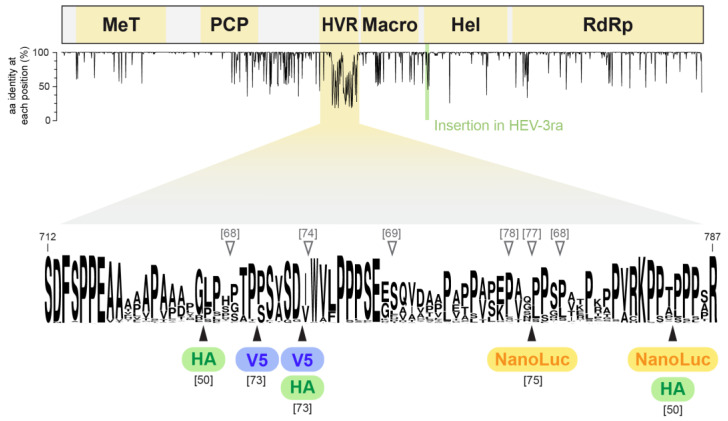
Engineered epitope tag and reporter insertions in HEV ORF1 genomic region. The sequence alignment of 155 HEV-3 ORF1 amino acid (aa) sequences, found in the databases, with exclusion of HEV-3ra and sequences harboring insertions in the HVR, served as basis for the aa identity sequence profile. Results of this alignment in the HVR, aa 712–787, is provided as an aa logo generated with the WebLogo application [76]. Some of the reported positions where natural insertions occurred, of which some served as reference for the targeted tagging approaches, are indicated by empty arrowheads with the corresponding citations [68,69,74,77,78]. Positions where epitope tags, HA or V5, as well as NanoLuc reporter have been successfully inserted in HEV-3 molecular clones, i.e., 83-2, p6 or pJE03-1760F, are indicated with their reference [50,73,75].

Current evidence points to the ORF1 region as having the greatest potential to accommodate reporter sequences. By contrast, the last third of the viral genome, encoding mainly ORF2, is likely more restrictive to heterologous sequence insertion given the tight and conserved protein–protein interactions required for capsid assembly. Using transposon-mediated random insertion and selection for infectious virus production, we, nevertheless, recently succeeded in identifying viable insertion sites in ORF2 (manuscript in preparation). In addition, Nishiyama et al. have recently reported a functional HEV genome expressing an ORF2 protein with a C-terminal FLAG tag [79]. Using the pJE03-1760F clone as a backbone, a FLAG tag sequence was fused to ORF2, followed by a stop codon and a duplication of the last 60 nt of the ORF2 sequence preceding the authentic stop codon. This sequence duplication circumvents disruption of a crucial cis-acting replication element present in the 3′ ORF2-coding region [40]. Importantly, quasi-enveloped virus-like particles harboring the FLAG-tagged capsid were secreted from transfected cells, as revealed by detergent treatment and immunoprecipitation analyses [79]. Whether these secreted particles as well as the intracellular virions are infectious remains to be clarified.

While recent studies showed that the HEV ORF1 and ORF2 proteins can tolerate the insertion of epitope tags and reporter genes, tagging of the ORF3 protein is likely more challenging, as almost its entire sequence overlaps with ORF2. More work is needed to further characterize and optimize the engineered tagged genomes. Overall, these hold promises for the study of virus entry, replication, and egress but validation of the versatility of the reporter systems as well as improvements in terms of virus production are a priority.

## 4. Challenges and Future Developments

The development of HEV in vitro infection and replication systems significantly contributed to a recent enhanced understanding of the viral life cycle, including entry, RNA replication, and virus production. Because current systems remain suboptimal, with modest efficacy and slow infectious virus production as well as rather low infectivity, there is not a single consensual HEV model system employed. Indeed, different cell lines, including some of hepatic origin, i.e., Huh-7 cells or sublines derived thereof, PLC/PRF/5, HepG2, and Hep293TT cells, but also A549 cells derived from lung tissue, are widely used to study HEV infection in vitro. Other systems, such as polarized stem cell-derived hepatocyte-like cells [80] or hepatic organoids [81] which are emerging, offer interesting alternatives to cancer cell lines but remain difficult to establish.

Among the infectious HEV clones, the Kernow-C1 p6 clone is the most widely used in cell culture because it produces the highest viral titers. However, it contains an S17 insertion in the HVR which provides a replication advantage, requiring validation of observations in an independent system. Efforts to establish new molecular clones, ideally with improved replication capacity, but also to identify highly permissive cell lines which may recapitulate all steps of the viral life cycle are crucial to the field.

Recent development in molecular biology, such as the CRISPR-Cas9 technology, now allow for the conducting of large genetic screening using pooled genome-wide single-guide RNA (sgRNA) library. This screening approach has proven over the last years to be extremely powerful for virus–host interaction study with numerous articles in the literature (among many others [29,82,83]), however, none concerned HEV. Beside genetic approaches, proteomics has shown to be very complementary to the study of virus–host interactions (reviewed in [84]). Together with improved sensitivity achieved lately with mass spectrometry (MS) detection methods, development of more robust HEV model systems shall serve the exploration of the host proteome involved during the complete viral life cycle.

Live-cell imaging experiments have not been conducted for HEV but may be an important step toward understanding of the leading mechanisms of infection. Engineering new recombinant viruses expressing fluorescently-labeled proteins may serve to follow the dynamic behavior of the HEV proteins during viral replication and, likely, the entire viral life cycle. Live-cell imaging may also include the detection of RNA molecules using insertion of targetable sequences in the viral genome (reviewed in [85]), as shown previously to study the translation and packaging of HIV-1 genome [86].

Reporter systems have proven to be instrumental in molecular virology to investigate viral life cycles, including the virus–host interactions. HEV model systems have undergone essential development in the last decade with the establishment of several infectious clones as well as selectable and reporter systems, which should translate into an improved understanding of the viral life cycle.

## Figures and Tables

**Figure 1 viruses-15-00869-f001:**
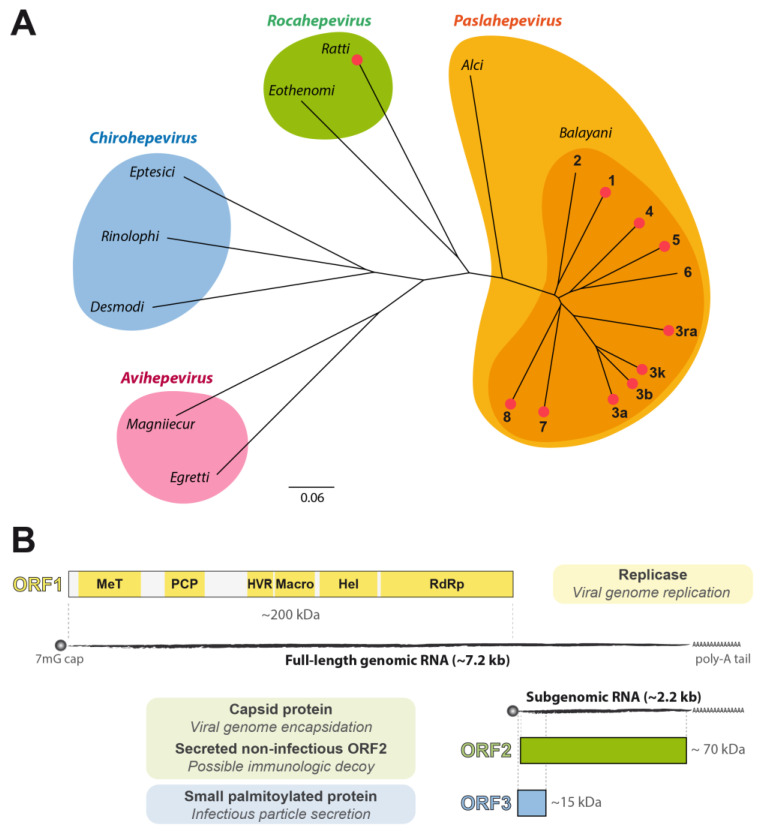
Phylogenetic relationship of the *Orthohepevirinae* and genomic organization of hepatitis E virus (HEV). (**A**) Sequences of full-length genomes from various viruses of the *Orthohepevirinae* subfamily were aligned with MUSCLE, followed by phylogenetic tree building using the neighbor-joining method (Geneious Prime software, Biomatters). The name of the viral strain and the corresponding accession numbers are the following: HEV-1, Sar55 (AF444002); HEV-2, Mexican strain (HPENSSP); HEV-3k, HEV83-2-27 (AB740232); HEV-3b, pJE03-1760F (LC126332); HEV-3a, Kernow-C1 (JQ679014); HEV-3ra, rbIM223LR (LC484431); HEV-4, TW6196E (HQ634346); HEV-5, JBOAR135-Shiz09 (AB573435); HEV-6, wbJOY_06 (AB602441); HEV-7, 180C (KJ496144); HEV-8, M2 (MH410176); *Paslahep. Alci*, AlgSwe2012 (KF951328); *Rocahep. ratti*, LA-B350 (KM516906); *Rocahep. eothenomi*, KS_10_1641/GER/2009 (MK192405); *Chirohep. rinolophi*, HEV/Shanxi2013 (KJ562187); Chirohep eptesici, BtHEVMd2350 (KX513953); *Chirohep. desmodi*, DesRot/Peru/API17_F_DrHEV (MW249011); *Avihep. magniiecur* (AY535004); and *Avihep. egretti*, kocsag02/2014/HUN (KX589065). Sequences for which a molecular clone is available are indicated by a red dot. The scale bar indicates the number of nucleotide substitutions per site. (**B**) HEV possesses a 7.2-kb positive-strand RNA genome with a 5′ 7-methylguanylate cap (m7G cap) and a 3′ polyadenylated tail (poly-A) composed of, at least, three open reading frames (ORF). ORF1 encodes the viral replicase composed of several functional domains, including a methyltransferase (MeT), a putative papain-like cysteine protease (PCP), a hypervariable region (HVR), a macro domain, an RNA helicase (Hel), and an RNA-dependent RNA polymerase (RdRp). During RNA replication, a subgenomic RNA species is generated and gives rise to the expression of ORF2 and ORF3 proteins, respectively, the viral capsid, produced in different forms (not represented here), and a small protein involved in virus secretion.

## Data Availability

Not applicable.

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
