# Peer review of "Expanding the Hepatitis E Virus Toolbox: Selectable Replicons and Recombinant Reporter Genomes"

_viruses, 2023, doi:10.3390/v15040869_

Round 1

Reviewer 1 Report

Well written review, thoughtfully organized and easy to follow; a very useful overview on the current state of the art of HEV molecular tools.

A few minor suggestions:

- lines 47-59: Please mention and discuss the function of the macrodomain; the HVR is not mentioned in the text at all, only in the figure

- Figure 1: at least in my copy, figure is grainy and pixilated. please make sure that final copy is high quality.

- schematic drawings of the different subgenomic replicons  and infectious clones might be helpful, depicting the different elements discussed in the text.

- a discussion on biosafety requirement of the infectious clones and trans-complementation systems would be helpful

-lines 246-248: verb missing in this sentence

Author Response

We thank the reviewer for the insightful comments and suggestions.

More specifically, we addressed points raised by the reviewer as follow:

1. The function of the Macro domain has now been introduced together with citation of a reference. Similarly, HVR is now described in the text as suggested.

2. The quality of Figure 1 has been checked. While we did not notice quality problem, on our side, we will pay attention to this in the next version of the manuscript.

3. The addition of a schematic representation of the different recombinant HEV constructs has been considered, already before our initial submission. However, we felt that such schemes would be poorly informative as there is almost no variation regarding the site of gene insertion for the replicon constructs (at ORF2 start codon) as well as for the infectious clones (within ORF1 HVR). Therefore we preferred to include a Table with size and type of inserted gene in replicon constructs, summarised in Table 1, and we prepared a Figure focused on ORF1 region for the infectious clones (Figure 2).

4. A sentence on the advantage of the trans-complementation systems to be experimentally manipulated at a lower biosafety level than infectious clones has been added in the text (Lines 206-207).

5. The addition of the missing word at the Line 246 has been done in the revised version of the manuscript.

Reviewer 2 Report

Dear authors, congratulations for your excellent work.

I would recommend the following minor changes:

1.    Line 34-37 should also include HEV-7.

2.    Line 38: Are there any convincing data indicating that HEV-2 leads to severe hepatitis with up to 25% fatality rate in pregnant women?

3.    Line 45: Not only HEV-3 but also HEV-4 may persist in immunocompromised patients and cause chronic hepatitis (PMID: 29432746; 30789146; 36275730).

4.    Line 95: (33208938; PMID: 22349148) may be Ma et al., Nature 588:308-314, 2020 and Carbajo-Lozoya et al., Virus Res 165: 112-117, 2012, respectively. They should be cited in the References section.

Author Response

We thank the reviewer for the insightful suggestions and the very positive evaluation of this manuscript.

More specifically, we addressed points raised by the reviewer as follow:

1. HEV-7 has been mentioned and a reference has been added.

2. There are recent data showing that mortality in pregnant women after infection with HEV-2 is very similar to the one observed with HEV-1 (Heemelaar S et al. Liver Int 2022;42:50-58.). The reference has now been added together with another review article summarising the epidemiology of HEV-1 and -2.

3. HEV-3ra, -4 and -7 as well as rat HEV have now been mentioned, together with HEV-3, as source of chronic infection. The EASL guidelines for Hepatitis E have been cited as reference.

4. The corresponding references have now been cited appropriately.